# Public perceptions of non-adherence to pandemic protection measures by self and others: A study of COVID-19 in the United Kingdom

Simon N. Williams[1,2]*, Christopher J. Armitage[3,4,5], Tova Tampe[6], Kimberly A. Dienes[3,7]

1 Centre for People and Organisation, School of Management, Swansea University, Swansea, Wales, United Kingdom, 2 Department of Medical Social Sciences, Feinberg School of Medicine, Northwestern University, Chicago, Illinois, United States of America, 3 Manchester Centre for Health Psychology, University of Manchester, Manchester, United Kingdom, 4 Manchester University NHS Foundation Trust, Manchester Academic Health Science Centre, Manchester, United Kingdom, 5 NIHR Greater Manchester Patient Safety Translational Research Centre, Manchester, United Kingdom, 6 Independent Public Health Consultant, Kassel, Germany, 7 Department of Psychology, School of Human and Health Sciences, Swansea University, Swansea, Wales, United Kingdom

* s.n.williams@swansea.ac.uk

**Data Availability Statement:** Although the authors cannot make their study's data publicly available at the time of publication, all authors commit to make the data underlying the findings described in this

## Abstract

### Background

Novel viral pandemics present significant challenges to global public health. Non-pharmaceutical interventions (e.g. social distancing) are an important means through which to control the transmission of such viruses. One of the key factors determining the effectiveness of such measures is the level of public adherence to them. Research to date has focused on quantitative exploration of adherence and non-adherence, with a relative lack of qualitative exploration of the reasons for non-adherence.

### Objective

To explore participants' perceptions of non-adherence to COVID-19 policy measures by self and others in the UK, focusing on perceived reasons for non-adherence.

### Methods

Qualitative study comprising 12 focus groups conducted via video-conferencing between 25th September and 13th November 2020. Participants were 51 UK residents aged 18 and above, reflecting a range of ages, genders and race/ethnicities. Data were analysed using a thematic approach.

### Results

Participants reported seeing an increase in non-adherence in others over the course of the pandemic. Reports of non-adherence in self were lower than reports of non-adherence in

study fully available without restriction to those who request the data, in compliance with the PLOS Data Availability policy. For data sets involving personally identifiable information or other sensitive data, data sharing is contingent on the data being handled appropriately by the data requester and in accordance with all applicable local requirements. Ethical restrictions related to participant confidentiality, as agreed during ethical review, prohibit the authors from making the data set publicly available. During the consent process, participants were explicitly guaranteed that the data would only be seen my members of the study team, including the authors of the current paper and any research assistants as approved via the ethical review. Data is kept securely and confidentially and anonymously on password-protected files for a minimum of ten years as per the ethical review requirements. Requests for access to data will be considered on a case-by-case basis, and data will be made available provided: a) any data made available does not compromise the identity or confidentiality of any participants and b) particularly where there are public health grounds for the release of data. For any discussions about the data set please contact the relevant members of the research team Dr Simon Williams (s.n.williams@swansea.ac.uk) or Dr Kimberly Dienes (k.dienes@swansea.ac.uk).

**Funding:** This research was supported by the Manchester Centre for Health Psychology based at the University of Manchester (£2000) and Swansea University's 'Greatest Need Fund' (£3000). The funders played no role in the study design, data collection and analysis, decision to publish, or preparation of the manuscript.

**Competing interests:** CJA is supported by NIHR Manchester Biomedical Research Centre and NIHR Greater Manchester Patient Safety Translational Research Centre. This does not alter our adherence to PLOS ONE policies on sharing data and materials." (as detailed online in our guide for authors http://journals.plos.org/plosone/s/competing-interests). The authors have no other relationships or activities that could appear to have influenced the submitted work.

others. Analysis revealed six main themes related to participants' reported reasons for non-adherence in self and others: (1) 'Alert fatigue' (where people find it difficult to follow, or switch off from, information about frequently changing rules or advice) (2) Inconsistent rules (3) Lack of trust in government (4) Learned Helplessness (5) Resistance and rebelliousness (6)The impact of vaccines on risk perception. Participants perceived a number of systemic failures (e.g. unclear policy, untrustworthy policymakers) to strongly contribute to two forms non-adherence—violations and errors.

## Conclusion

Findings suggest that latent and systemic failures—in the form of policy decisions that are commonly experienced as too changeable, inconsistent and confusing, and policy makers that are commonly perceived as untrustworthy–may play a significant role in creating the conditions that enable or encourage non-adherence.

## Introduction

The COVID-19 coronavirus (COVID-19) pandemic has arguably presented the most significant challenge to global public health in a century [1]. Non-pharmaceutical interventions, including social distancing measures, are an important means through which to control the transmission of such viruses. During the pandemic, in the UK, the social distancing measures that are the focus of this study included, but were not limited to: keeping physically separate (1–2 meters in the UK), only meeting with others in ways permitted under current legislation, and self-isolating when required to do so [2].

One of the key factors determining the effectiveness of such measures is the level of public adherence to them. As a pandemic draws on, is important to explore whether, and in what ways, the longer duration of measures may impact adherence and also the reasons for non-adherence amongst those who are not fully adhering to them. There has been much discussion in the media about whether 'behavioural fatigue' may potentially lead to increases in non-adherence [3]. However, this concept is controversial amongst academics, partly because and the term behavioural fatigue has been vaguely applied and ill-defined [4], and partly because of the lack of empirical evidence to support the fact that there is significant 'behavioural fatigue' around adherence [4]. In the UK, longitudinal public surveys generally note that adherence has been overall high over the course of the pandemic [5, 6]. However, researchers reporting on the survey data caution that there are high levels of self-reported confusion or a lack of understanding of guidelines [5]. Also, reported 'complete adherence' (where people report following all rules with no bending or even minor infringements) is considerably lower than reported 'majority adherence' (where people are following most but not all rules) [5]. Some rules (e.g. observing the 2-metre rule or visiting extended family when not permitted to do so) may be being flouted more than others (e.g. not self-isolating when advised to do so) [6]. However, a recent international study posited that 'fatigue' might explain an observed decline in adherence for 'high-cost' behaviours like social distancing (but not for 'low-cost' ones like mask-wearing). For high and middle income countries this decline was reversed later in the pandemic [7]. Further behavioural research is needed to add to our understanding of the potential existence and role of "fatigue" during the pandemic [8].

There is a rapidly growing body of research exploring factors associated with adherence and non-adherence to COVID-19 measures. Surveys of public behaviours suggests that some of the main factors contributing to high adherence include: high perception of risk of contracting COVID-19 [9–11], perceived social support [12], seeing others' adhere [13], believing that you have already had COVID-19 [14], greater accessing of health-related information [15], trust in government [13] and political ideology [16]. Also, certain demographic factors have been found to predict high adherence, including: older age [5, 11] female gender [9], higher education level [9, 15]. Of course, adherence levels can vary across countries [15], due to a variety of reasons, including levels of pre-existing trust in authorities and the extent to which a country has a 'tight' or collective culture [17].

Much of this research on adherence to coronavirus measures has taken the form of quantitative surveys, which may give us an understanding of the extent of adherence and related factors, but often fall short on understanding the 'why' associated with such adherence. Qualitative research on experiences and perceptions of adherence can complement existing quantitative research by exploring in-depth some of the reasons behind instances of non-adherence, from the perspectives of the participants themselves, which can then be used to adapt policies to maximize impact. In this paper, we present results from an ongoing longitudinal qualitative research study looking at public experiences of, and attitudes to, COVID-19 measures put in place to reduce the spread of the virus at a population level [18, 19]. We focus on adherence to government mandates and regulation–particularly including social distancing measures such as keeping 1–2 metres apart from others, only meeting and mixing with others where and when permitted, but also including other mandated measures such as facemask requirements. In a previous paper, we explored experiences and perceptions of self-adherence and of adherence in others to social distancing measures [18]. We found that the majority of our participants reported high adherence to measures for themselves, but that reports of non-adherence in others were high [18].

In addition to the rapidly emerging empirical literature on COVID-19, we also drew on behavioural theory around rule-following and breaking, in particular Reason's [20, 21] theory of human error. According to Reason [20], violations are deliberate deviations from safe practices, behaviours, standards or rules, whereas errors are the failed actions to achieve their desired goal. Errors include 'slips' and 'lapses' (failures of execution, e.g. through forgetting or misunderstanding information) and 'mistakes' (failures of intention, e.g. ineffective plans of action to achieve a desired goal) [20]. Additionally, Reason [22] distinguishes between two domains of failure: active and latent failure. Active failures refer to the unsafe acts (i.e. violations and errors) at the individual (person) level, whereas latent failures refer to the *conditions* under which such unsafe acts are enabled or not prevented (i.e. 'error-provoking conditions') at the institutional (systems) level [22]. Latent failures arise from decisions made by those designing, writing, managing, implementing and enforcing (safety) policies and procedures [21]. Deliberate rule violation versus errors in following rules are two very different forms of non-adherence, but literature on COVID-19 behaviour is yet to sufficiently distinguish between different types of non-adherence. Reason's model enabled us to separate out the intention behind the behaviour into violations vs. errors. The present paper explores two main research questions: (1) What are participants' perceptions of the extent of non-adherence to COVID-19 measures in self and others to COVID-19? (2) What do participants' feel are the main reasons for non-adherence to COVID-19 measures, in self and others? The paper explores these questions in relation to behavioural theory including Reason's model of human error [20–22]. The study therefore aims to contribute to existing knowledge around people's experiences and perceptions of non-adherence to COVID-19 measures, with a view to contributing to the evidence base for the current pandemic as well as future infectious disease outbreaks.

## Materials and methods

Data from this study came from 12 focus groups with 51 participants were conducted between 25th September and 13th November 2020. Participants were all UK-based adults aged 18 years and above. The study period included a number of important policy developments, which varied across the for UK nations, that provide a context for perceptions of government mandated rules including: the introduction of local lockdowns in Wales (introduced on 27 September); Scotland's 16 day ban on drinking alcohol in licenses premises (introduced on 7th October); the 'three tier' system of COVID-19 restrictions in England (introduced on 12th October); the 4 week closure of pubs and restaurants in Northern Ireland (introduced on 14th October); the 19 day 'firebreak' lockdown in Wales (introduced 23rd October); Scotland's five-tier system (introduced 21st October); and England's four-week national lockdown (introduced 5th November). Beyond these major developments, a full list and description of all the specific policy changes, including how they differed across the four UK nations is beyond the scope of this research, but full details and ongoing updates of pandemic policies can be found on various 'policy trackers' [23] (e.g. https://www.bsg.ox.ac.uk/research/research-projects/covid-19-government-response-tracker).

### Participants and data collection

The methodology used in this study has been discussed in previous publications [18, 19]. Online focus groups were a necessary requirement of the pandemic, but have previously been shown to have the advantage of conveniently obtaining the views of a diverse group of geographically disparate group of individuals (although they potentially preclude those without the requisite communication technology) [24, 25]. Purposive sampling was used to obtain as diverse a sample as possible, including gender, age and race and ethnicity and geography (across the UK) (Table 1) Participants in the final sample were recruited from across England,

**Table 1. Demographic details reported by participants.**

| Characteristic | N |
|---|---|
| *Gender* | |
| Female | 25 |
| Male | 26 |
| *Age range* | |
| 20–29 | 9 |
| 30–39 | 12 |
| 40–49 | 18 |
| 50+ | 5 |
| Undisclosed | 7 |
| *Ethnicity* | |
| White British | 32 |
| Asian or Asian British | 9 |
| Black or Black British | 4 |
| Other | 6 |
| *Country* | |
| England | 32 |
| Wales | 17 |
| Scotland | 1 |
| Northern Ireland | 1 |

Wales, Scotland and Northern Ireland, although the final sample was primarily from England and Wales (primarily an effect of the snowball sampling technique used). Recruitment took place via a combination of social media advertisements (targeted Facebook ads), online advertising via local community and volunteering sites (e.g. Facebook groups (local community interest groups), Gumtree volunteering groups) and social media snowball recruitment (e.g. via Twitter). Volunteering and community sites were chosen due to their common use as sites to advertise research studies (although this increases the potential risk of a biased sample-toward those who may be more inclined to take part in research studies). Although targeted social media ads attempted to recruit additional members of Black and Asian Minority Ethnic (BAME) participants and those aged over 50, the final sample included a large proportion of white participants aged under 50. As part of the initial recruitment, participants were told the study was looking at the UK public's perceptions and experiences of social distancing measures during the pandemic.

Each focus group (average 4 participants per group) met virtually via a web videoconferencing platform (Zoom) for approximately one hour. Participants joined using both video and audio. Focus groups were organised and moderated by SW and KD. The topic guide for the focus groups was initially developed using existing literature on adherence to health behaviours (discussed above) as well as rapidly emerging research on COVID-19 public attitudes. Focus groups were semi-structured and contained a lengthy section on adherence, particularly focused on adherence to government mandates and legal and official measures around social distancing (as noted above, including keeping 1–2 metres physically distant, only meeting and mixing with others as and when permitted). In discussing social distancing measures, participants at times voluntarily discussed (non-)adherence to other rules, including mask mandates. The main topics for the focus groups included: what people thought about their own adherence and the adherence of others to COVID-19 social distancing measures, what they thought the reasons for adherence or non-adherence were, as well as broader topics related to people's views and experiences around any impact of the pandemic on work life, social life and mental health–the focus of previous and future publications from the project. The focus group schedule is included as S1 File.

Ethical approval was received by Swansea University's School of Management Research Ethics Committee. All participants gave informed consent, both written and verbal. All data were kept securely an confidentially in line with ethical requirements, and where data is presented below, all quotes are anonymised to protect participants' identities.

## Analysis

Data were analysed in accordance with a thematic approach as described in Coffey and Atkinson [26]. We took an iterative, pragmatic approach to data collection and analysis, wherein emergent themes from each focus group were used to add to or refine questions during subsequent focus groups. All Zoom focus groups were audio recorded and transcribed. SW and KD analysed the transcripts and, in discussion with CA and TT, developed and applied the thematic coding framework. We also examined the data to look for information which might not 'fit' with the emerging themes (i.e. negative case analysis) the data [27]. These are discussed under 'alternative accounts' in the Results section. Initial research questions informed the analysis process, which proceeded abductively [26], making links between codes and concepts (including concepts derived from existing research), in order to generate the themes discussed below [26]. Data collection and analysis continued until no new significant themes were developed. Data were analysed in NVivo (V.11.4.3, QSR).

## Results

### Non-adherence in self and others

Participants' accounts were found to relate to two forms of non-adherence (or 'aberrant behaviour'): *violations* and *errors* [20, 21]. Whereas participants tended to discuss their own non-adherence, where it occurred, as a type of *error* (i.e. a (frequently unwitting or non-deliberate) failure of planned actions to achieve their intended consequences), they tended to discuss others non-adherence as types of *violation* (i.e. a frequently conscious or deliberate deviation from standard and 'safe' practices (cf. Reason [20, 22]).

Overall, as was observed earlier in the pandemic [18], reports of non-adherence in self were lower than reports of non-adherence in others. Most participants felt that adherence to COVID-19 measures observed in others was currently lower than it has been earlier in the pandemic. In particular, participants reported seeing others failing to stay 1–2 metres apart (especially in shops) and knowing of others meeting with family and friends in households (in ways not permitted at the time). Reports of non- adherence in self were more prominent at this stage of the pandemic as compared to reports of non- adherence in self at an early stage of the pandemic [18]. However, participants described trying to follow the measures as best they could but expressed how doing so had often been challenging (these reasons for which are discussed in more detail below). Whereas non-adherence in self tended to be described in terms of these situational factors (challenges), non-adherence in others tended to be described in terms of personality factors (therefore relating to the fundamental attribution error–specifically the actor-observer bias [28], as discussed in more detail below).

These broad findings concerning violations and errors in self and others will now be discussed in more detail in relation to the specific reasons for non-adherence that emerged from the analysis.

### Reasons for non-adherence

Analysis revealed six main themes related to participants' reported reasons for non-adherence, where it was being observed: (1) Alert fatigue (2) Inconsistent rules (3) Lack of trust in government (4) Helplessness (5) Resistance and rebelliousness (6) Reduced perception of risk and the prospect of a vaccine. We did not find any obvious patterns or differences according to the demographics of the participants, with a mix of genders, ages and races and ethnicities being represented in each theme. As noted above, regardless of their demographics, all participants reported generally high self- adherence. In this section we describe these themes, before further exploring their interrelations and implications in the discussion section.

**Alert fatigue.** One of the main reasons for non-adherence, particularly non-adherence in self, was the high volume and frequency of information that they had been exposed to over the course of the pandemic. Specifically, the frequent government announcements related to what was often perceived to be constantly changing and complex rules left many feeling "lost" (Participant 7, Female, 20s) or "confused" (Participant 31, Male, 40s), and that it was "impossible to keep up with the rules" (Participant 29, Female, 40s). As one participant put it, "I'm feeling a bit fatigued, sick of it" (Participant 13, Male, 30s). This constant stream of information could be seen to be causing information fatigue, or rather a particular form of 'information fatigue' referred to as 'alert fatigue' [29]. This concept is derived from research on clinical decision-making support systems, and its understood as the mental state that comes from receiving too many alerts that consume time and energy which can cause important alerts to be ignored along with clinically unimportant ones [30, 31]. Alert fatigue was observed where participants felt it was difficult to follow ("keep up with") or remember what the rules were. This in turn

meant that potentially important information was being missed or not taken as seriously as it might or should have, potentially leading to adherence 'errors':

"It's been hard this lockdown, with all these government measures and restrictions. . . . I don't know what the new legislation is. It's hard to keep up-to-date with all of this information. I've come to the decision where I am going through day-by-day but placing less attention on whatever the government is saying. . .. I'm not as concerned about following the measures as seriously as I should have done." (Participant 2, Male, 30s)

Whereas government announcements ("conferences") had once been given significant attention and been taken seriously by the public, the cumulative effect of the over-exposure to these alerts was leading participants to become desensitised or habituated to this information. Participants felt that the constant exposure to new information on the pandemic, meant that it became hard to distinguish between important announcements about new rules that directly affected them, and less important or superfluous information ("just updating"):

"It does feel like they are changing the rules like every week, and then people don't take them seriously, and I don't blame them . . . I go on Facebook and it's coming up every few days as a notification 'Boris [Johnson, UK Prime Minister] is live' . . . I never watch them anymore because I don't know if they are going to be new rules or whether he is just updating us, I don't know what's going on . . . If we just had fewer and they concentrated just on the rules, individuals might concentrate more and take it more seriously." (Participant 8, Female, 20s)

The sense of alert fatigue over the frequency of announcements and rule changes was compounded for some participants by the perceived lack of clarity in messages, partly because of the overload of information *per se*, and partly because much of it was technical information that was, in their view, insufficiently explained or translated:

"I watch the [Welsh Government] announcements on Facebook Live stream, and after he [Mark Drakeford, Welsh First Minister] talks, it just seems like a Q and A for an hour, with all these long words, and you haven't got any further with what the rules are, and you are left feeling 'what does that mean?'" (Participant 39, Female, 20s)

**Inconsistent rules.** In addition to feeling over-exposed to information about COVID-19 measures, participants also expressed feeling as though measures were often "confusing because of the mixed messages" (Participant 3, Male, 20s). This was discussed both in relation to non-adherence in self and in others. One of the main causes of this confusion was the perception that rules were inconsistent, either because they were changing so much over time (as discussed above as a cause of alert fatigue) or because they were inconsistent across place–that is between countries in the UK and between different regions within each country, due to the multiple policy changes noted above and in policy trackers [23]). Participants criticised what they saw as the inability, or unwillingness, of political leaders to create consistent policy and present a unified front across the different countries:

"I'm most upset with our individual home country leaders. The fact that they cannot get on the same page, the rules are already confusing, but then they make them more confusing . . . they should have been able to come up with one set of rules, it's absolutely insane" (Participant 13, Male, 30s).

Such criticisms of inconsistent rules might be viewed as criticisms of latent failures (i.e. inadequate policymaking). Participants also expressed concern that the lack of consistency would also be a problem for the future:

"I don't feel like anybody is taking anything very seriously, because one rule is being made and another one contradicts it . . . especially with these new tiers . . . People aren't really aware of what they are doing or why they are doing it. . . Everywhere has adopted the rules, but then they have refined them, changed them and interpreted them to suit themselves. . . .. And so, is that our life now? That we have to wait for different rules, like every month, rules that apply to some and rules that don't apply to others" (Participant 28, 50+)

Non-adherence was generally seen to stem from the lack of clarity and consistency in the way in which rules were made and communicated. However, although non-adherence in others was frequently framed as violations–here as a deliberate exploitation of the inconsistent rules ("interpreted them to suit themselves")—non-adherence in self tended to be more so framed as an unintentional, and perhaps unavoidable errors ("I try to abide by them"), consequence of the inability to make sense of, the rules:

"I'm not some total non-conformist. Yes, the rules are there to help, and I try to abide by them, but it's just gone to the extreme now where, if I asked the three of you now, what the rules are, we would probably all say something different." (Participant 6, Male, 20s)

**Lack of trust in government.** Participants also discussed a general lack of trust in, or "respect" for, the UK government, who were seen to have "handled the pandemic very badly" (Participant 29, Female, 40s). The lack of trust in government was seen as something that also led to the type of rule violation (rule-exploitation) discussed above. It was felt that a lack of respect in authority permitted people to "make up" their own rules:

"We don't respect our government, we don't respect their rules. We might follow something that we decide to follow, but then once we are allowed to say 'oh yeah but its Christmas' or 'oh but somebody's getting married' and 'we need to do this, we need to do that', then everybody makes up their own rules and goes 'well we are going to get away with it anyway'." (Participant 28, 50+)

Some participants argued that the lack of respect for government was being used by some to account for engaging in forms of social mixing that was not permitted within the rules. As the above quote illustrates, rules were often seen to contain exceptions or loopholes that could be exploited (i.e. "make up their own rules"), something that negatively impacted motivation to adhere or permitted people to subjectively interpret the rules to permit or justify desired behaviour (e.g. meeting more people than might be permitted in a given context). One key reason for the loss of trust was the well-publicised instances in which politicians were seen to be subjectively interpreting the rules to their own benefit. Participants argued that if those in positions of authority were unable or unwilling to follow rules, why should the public be expected to do so? This lack of trust was perceived to be exacerbating the stress people were experiencing during the pandemic:

"There is no doubt that the way the government is communicating and handling things is going to add to people's stress. . . . [I]t's happening now . . . it doesn't help that the politicians are interpreting the rules to their own benefit as well." (Participant 31, Male, 40s)

**Learned helplessness.** Another major theme related to non-adherence in both self and others was a feeling of learned helplessness (being "fed up" and "giving up"). The learned helplessness theory originated from the discovery that a certain type of behaviour resulted from perceived helplessness, including repeated attempts to change the situation followed by giving up and becoming almost catatonic [32]. Abramson and colleagues [33] stated that we attribute our helplessness to a cause (stable/unstable, global/specific, consistent/inconsistent) and that stable, global and inconsistent causes of helplessness lead to depression. Participants attributions of the of the pandemic appeared to be stable (not going away), global (pandemic), and inconsistent (see above alert fatigue and inconsistent rules). Learned helplessness was in some respects a product of, or response to, the alert fatigue, inconsistent rules, and lack of trust in government. For example, some linked their feeling of helplessness, and the associated emotional toll (feeling "down") to the over-exposure to news on COVID-19, including the frequently changing rules, and the frequent news coverage them, discussed above (alert fatigue): "I have just given up. I don't even look at the news, I mean it changes every day and I'm fed up. It just gets you down really" (Participant 29, Female 40s). Some participants reported "struggling to have a positive outlook on life" (Participant 6, Male, 20s) and generally feeling pessimistic over the future, particularly in regard to how long they would be subject to some form of coronavirus measures. This included looking ahead to upcoming events that would normally be happy, but which would be different this year:

"We are going into the New Year, and normally we have that feeling of setting new goals and having new things to look forward to. What is the mental state of people going to be? . . . [P]eople are going to have so much uncertainty about the new year and Covid." (Participant 25, Male, 40s)

Participants tended to attribute people's sense of helplessness to a loss of control over their lives in keeping with loss of agency in learned helplessness theory: "People have lost control over their life. . . and when you lose control over your life, that's it." (Participant 15, Male 40s) For some, the control had been taken from them by government regulation (external attribution). The external attribution of the cause of helplessness can lead more to anger or paranoia than depressed mood [33], and participants expressed a sense of resentment for the way in which, as they perceived it, general civil liberties were being "taken away" (for example, the freedom to socially interact in ways of their choosing), at the same time as "small" freedoms were being "given back":

"I am not happy that they take things away from you and then find little ways of giving bits of it back . . . it's getting very weird, and I wish they would be more truthful and just said, don't mix with anybody, use your common sense." (Participant 28, 50+)

For a number of participants, the loss of autonomy and control over their lives ("they take things away from you"), compounded by the inconsistency ("giving bits of it back") and the lack of trust in government ("I wish they would be more truthful"), led them to follow (or advocate for) "common sense", as opposed to official rules.

Learned helplessness, including a lack of control or optimism for the future, had been accumulating over the course of the pandemic. One of the main factors involved was the reported constant uncertainty surrounding whether measures would change, and ultimately how long some combination of measures would be in place (unstable attribution). The feeling of there being no defined end point ("is this our life now?") added to the sense of helplessness. Participants linked this uncertainty and the growing feeling of helplessness to observations of non-

adherence, of "giving up" trying to strictly follow the rules ("we might as well do what we want"):

> "I think at the beginning [of the pandemic], we were told it was three weeks, which then turned to three months, which now has turned to six months, and this is so past what anyone could have expected, everyone has thrown in the towel, like 'we don't know when it's going to end' . . . I think people have lost hope and just half accepted how this is and are going back to a normal life as best they can instead of being really strict and taking it seriously." (Participant 8, Female, 20s)

**Resistance and rebelliousness.**    Another theme that emerged as a reason for non-adherence, particularly non-adherence in others, was people's resistance and rebelliousness in relation to COVID-19 measures. Often this was seen to take the form of deliberate rule violation. Participants viewed it as a consequence of, or response to, the feeling of being "fed up" or a loss of motivation to continue to adhere:

> "I'm hearing stories of people being rebellious. . .. there is a general feeling of being fed up. . . there is a lady here who would be outspoken about the rules, if people were breaking them, but even she was last night saying she was away with her daughter so now even she is breaking them." (Participant 26, Female, 40s)

Participants felt this would worsen in the future, particularly around times that were expected to be happy and involving togetherness:

> "I think a lot of people . . . are going to break the rules, even if it is slightly. I think if you have got a family, and people are told you can't mix with another household unless it is outside, people are going to say 'Its Christmas and I want to see my family". People are going to say I don't care, I want to have something to look forward to." (Participant 30, Male 40s)

For some, this rule-violation was a form of protest to the loss of autonomy and control ("we don't like following rules") and the inconsistent rules ("the mixed messages"), and was something that would worsen in the future, potentially even taking the form of "civil unrest" or "protest" (Participant 49, 50+):

> "[I]t is getting on for nearly a year . . . how long can you go without seeing people? I'll go to visit [family's town]. People are going to break the rules. . . It is going to get even worse, where people are going to get completely fed up . . .. We are quite bolshy in the UK. We like a riot and don't like following rules . . . there is going to be civil unrest, because there is already in other countries, and it's because of the mixed messages." (Participant 29, Female 40s)

For some, resistance was the attempt to re-exert control over their lives, to reassert their autonomy, by focusing on controlling what they could control. In some instances, this included overt rule-breaking (e.g. not wearing masks when required):

> "The more you know you can't really change the outcome of it, all you can do is make the most of your own life and adapting to it. I feel very overwhelmed . . . I don't want to talk to people because I don't want to get into an argument about it . . . I'm not wearing a mask

and I'm not going to. I'm going to stick to what I believe . . . I am responsible for my life and my daughter." (Participant 26, Female, 40s)

Others felt that resistance would take the form of rule-exploitation (strategic interpretation of the rules to suit their interests). They also however framed non-adherence as a response to the loss of autonomy and control; something that would worsen in the future,

"[P]eople, fair enough, will not stand for being controlled. People will want to travel and see their loved ones . . . If we are in a tiered system people will try and interpret the rules in their own way . . . we are not great at being dictated to, or doing things that are for our benefit. . . and it's going to get worse from a civil unrest point of view. I fear we are sleepwalking into a police state." (Participant 31, Male, 40s)

Many participants discussed examples of ways in which people were engaging in small, creative and subtle forms of rebellion to government rules. These forms of rule violation included engaging in social rituals now perceived as deviant (e.g. handshaking, not wearing masks when required to), or mis-appropriating new rituals (e.g. improperly wearing masks) as more subtle violations (through passive resistance to rules):

"You have got some people who are like 'Covid is here'; you are seeing the masks, but then you have got this other spectrum where it is almost like a rebellion against Covid . . . I had a guy at the burger van the other day, and he goes 'alright [Name], how are you?' And he goes to shake my hand'. And I said 'er, social distancing?' And he goes '*I don't believe in that shit*' and he just grabbed my hand and shook it . . . And I was like 'what are you doing, now I'm going to have to wash my hand!'" (Participant 41, Male, 30s)

**Reduced perception of risk: Anticipation of vaccinations.** As discussed in the methods section above, one benefit of conducting qualitative research in real-time during the pandemic is the opportunity to capture emerging themes in a rapidly evolving policy and scientific landscape. One example of this was the announcement on the 9th November 2020, that a leading vaccine candidate (developed by the pharmaceutical and research companies Pfizer and BioNTech) had announced Phase 3 clinical trial results which indicated that their COVID-19 vaccine was 90% effective in preventing the disease, and might start being rolled out as early as the end of the year [34]. In the five focus groups conducted after the announcement, an additional theme which emerged was the participants' perceptions of how the prospect of a vaccine might serve to reduce the perceived risk, or threat, of COVID-19, and how this might adversely impact public adherence to measures. Views coded under this theme tended to focus on the fact that the announcement made the prospect of a vaccine more tangible, and that the pandemic was losing its "fear factor" (Participant 13, Male 30s). This theme was nearly always focused on adherence in others (with no participants reporting that it would make them less likely to adhere). Some participants felt this would have a general impact on adherence:

"I get the impression people are going to be more relaxed, like 'oh there is a vaccine now, the problem is fixed. I went to Tesco twice today and both times I saw people without masks for the first time [since it was made compulsory]. I personally think it [the vaccine] takes the danger away." (Participant 38, Male, 30s):

"I think people will be more blasé now, or people will think 'we can carry on now because we have a vaccine, we will be fine.'" (Participant 46, 50+)

This could be compared to arguments earlier in the pandemic around mask-wearing, where concerns existed that mask wearing might encourage a false sense of security thereby discouraging people from maintaining good hygiene or social distancing behaviours [35]. Other participants felt that news would only impact the adherence of those who were already prone to non-adherence (i.e. it would serve to justify and increase their non-adherence):

"I think of the people who are currently feeling that way inclined, you know to bend the rules, break the rules, stretch the rules, they probably will break the rules more. However, I wouldn't say that about everybody. There are groups who are still taking things very seriously, still observing the guidance and just because the vaccine has been announced I don't think people are going to go and stop. I personally feel like we don't have enough information to know if it is going to be the golden bullet . . . But those who are inclined to break the rules are more likely to take the news of a vaccine to mean 'it's all over let's crack on with life.'" (Participant 34, Female, 30s).

## Alternative accounts on (non-)adherence

Although the themes discussed above constituted the prevailing views expressed during the focus groups, negative case analysis suggested that some participants held different related to adherence and non-adherence. For example, although most participants argued that adherence in general was waning over time, some participants offered alternative accounts. Some argued that adherence to social distancing had been high during the first lockdown, had lessened during the summer, but was now being "taken more seriously again" (Participant 40, 50 +). They attributed this to various factors, including most notably the perception of risk had fallen and then risen again corresponding to the number of cases between waves. Some participants described how for them personally, adherence hadn't changed much over the course of the pandemic. These participants tended to characterise themselves as being quite strict or conscientious about the rules and as such felt that they hadn't changed their behaviour (e.g. "we never went fully out, we never went to a restaurant" (Participant 11, Female 40s) during the summer when measures had relaxed. Also, not all participants felt that policy had been inconsistent, hard to understand or follow. For example, some participants "liked" the tier system because it helped them better "understand what risk we are at in relation to those around us" (Participant 10, Male, 30s).

## Discussion

Six themes emerged from our qualitative analysis across all focus groups: Alert fatigue; inconsistent rules; lack of trust in government; learned helplessness; resistance and rebelliousness; and (in later focus groups) reduced perception of risk due to the prospect of a vaccine. These themes are not mutually exclusive and are inter-related in complex ways. For example, in some instances the feeling of helplessness was attributed to the frustration of not being able to "keep up" with rules (alert fatigue) or the feeling they were inconsistent or didn't "make sense". Also, the resistance and rebelliousness reported was in part seen as a response to the frustration over inconsistent policy and a lack of trust in or respect for government, compounded by the perceived loss of agency and control (attributions linked to learned helplessness. Taken together these factors were associated with reduced adherence to rules either in the form of violations or errors. However, a key finding is how prominently *latent conditions* figured in participants accounts of non-adherence. Rather than violations and errors to stem from individual failings, they were instead seen to stem from systemic (latent) failures [20],

including what were perceived as overly frequent, confusing and inconsistent rules set by what were seen to be untrustworthy decision-makers.

Recent survey evidence suggests that self-reported adherence (or at least 'majority adherence') in the UK has generally remained high and stable throughout the pandemic [5]. It may be the case that discrepancies exist between people's perceptions of (high) adherence in self and their perceptions of (low) adherence in others. However, our findings suggest that, from the perspective of our participants, non-adherence in both self and others may be more apparent, as compared to earlier stages in the pandemic [18]. All six themes were observed in related to both self and other non-adherence. However, certain themes were more prominent in relation to participants' accounts of non-adherence in self compared to non-adherence in others (and vice-versa). For example, participants were more likely to discuss non-adherence in self in terms of alert fatigue or subjective rule interpretation and others' non-adherence in terms of resistance and rebelliousness and rule violations. A number of factors could account for this discrepancy, including social desirability bias (the desire to appear conscientious) or selection bias (those more likely to adhere might also be more likely to take part in a research study on adherence and might be more likely to be recruited via non-random sampling via social media). Perhaps one explanation for this is the fundamental attribution error, or specifically the actor-observer bias [27], which the holds that people tend to believe others' actions stem from stable characteristics (e.g. rebelliousness) while seeing their own actions to also stem from contextual factors (e.g. an overload of complex information). It is important to note however that inconsistent policy and a lack of trust was pervasive in both self and others' non-adherence.

Although, as discussed above, the notion of 'behavioural fatigue' is a largely vague and poorly evidenced concept [3], our findings do suggest that a very specific form of fatigue—alert fatigue—may be an important factor for non-adherence to guidelines. This phenomenon is explained as a result of cognitive overload (where too much information cannot be processed, or retained, effectively) or desensitisation (where repeated exposure to alerts leads to declining responsiveness to them) [36]. In their study of public health care providers, Baseman et al. [37] argued that "during a pandemic when numerous messaged are sent, alert fatigue may impact ability to recall when a specific message has been received due to the 'noise' created by the higher number of messages". Our findings suggest that in the context of the current coronavirus pandemic, alert fatigue is a wider phenomenon being experienced by some within the general public. Indeed, the concept of alert fatigue has its roots in the much broader concept of "information overload" which was originally taken to be where a person develops a "blasé outlook" as a means to cope with a constant influx of stimuli [38]. Many of our participants described how they were no longer paying as much attention to—or even actively avoiding—what they perceived as "constant" news on COVID-19, including what was perceived as excessively frequent and detailed announcements (alerts) made by political leaders. As such, in their desire to provide "transparency", and to communicate "the science" behind policy, government may have counter-productively "overloaded" the public with information to the point at which some are actively avoiding it. Such avoidance could be construed as a coping mechanism both in regard to the cognitive overload (of having to filter through the "noise") and in regard to the anxiety or stress caused by the focus on the pandemic and the measures.

Of course, the frequency of the alerts was also related to the fact that policies were also changing frequently. Our findings suggest that participants were often confused over what they perceived to be constantly changing rules, which were sometimes inconsistent or didn't "make sense" to them. Significantly, confusion was constructed not as an individual error but rather as a consequence of the latent failures or error-provoking *conditions* [20] created by confusing policy. The differing policy approaches and timelines between the four nations was

a key source of confusion, as was the introduction of local lockdowns and, in England, the tier system. Significantly, the geographic variability, in addition to the variability over time, was perceived by some as being a reason as to why they were no longer able to, or no longer chose to, follow "the rules", and can as such be seen as an example of a latent or system failure.

Related to both alert fatigue and inconsistency in rules was a general sense of a lack of trust in, or respect for, government. The relatively low confidence in the UK government has been documented in longitudinal surveys [5, 39]. Additionally, it is well-established that role modelling by people in positions of leadership is an important factor in motivating adherence, and research on COVID-19 is starting to explore the adverse impact of poor behavioural role modelling (the so-called "Cummings Effect" [40]). As seminal research in psychology has shown, trust and confidence in those in authority plays a key role in the extent to which they are followed [41], as does positive role modelling [42]. Milgram [41] found that polyphony—that is the disagreement between conflicting voices of authority—negatively impacts compliance by increasing the likelihood that people will have to decide for themselves what action to take. In the case of COVID-19 measures, there has been an increasing trend towards polyphony amongst political leaders across the UK, something participants noted when contrasting recent measures to the initial UK-wide lockdown in March. The existence of different announcements about different countries and regions coming from multiple voices (e.g. national and regional political leaders) likely compounds the general sense of alert fatigue caused by the frequency of announcements. Our findings suggest that, from the perspective of our participants, a more unified set of measures across the UK, or at least the appearance of a more collaborative and unified voice, would have resulted in an improved capacity to understand, and thus potentially follow, rules.

Large UK surveys captured increases in depression and anxiety during the first wave of the pandemic, that largely improved over the summer, and worsened again during the second wave [5]. Our previous qualitative research, conducted during the first wave of the pandemic, supported these findings and explored people's experiences of social isolation. We found that social distancing was already leading to feelings of 'loss'. Practical losses, (e.g. social interaction; income; structure and routine) and psychological losses (motivation and self-esteem) which in turn were having negative impacts on mental health (increased depression and anxiety [18]. Learned helplessness theory proposes that repeated loss of control or agency over time can lead to depressed mood [32]. Learned helplessness in our study appears to have emerged as a result of the sustained social sacrifice and social suffering (a sustained sense of loss), coupled with a growing sense of a lack of control, as the pandemic has drawn on. Participants reported perceiving the pandemic as stable (something that would never end or cease to have an influence) and uncontrollable (the increasing levels of the pandemic despite sacrifices being made). Participants tended to make external attributions, seeing the government as causing their loss of control due to inconsistent measures and lack of clarity.

Participants were, in a sense, being 'conditioned' to feel as though they had little control over the outcome of the pandemic and over their lives in general, and (drawing a comparison to seminal experimental studies [43] they had been experiencing a 'failure to escape' social restrictions loss of freedoms). Similar to 'giving up' in the context of health behaviours, such as giving up trying to stop smoking after repeated failures, repeated failure in the context of coronavirus measures meant 'giving up' trying to adhere to the rules. Such learned helplessness stems from cognitive, affective and motivational deficits [44] which some of our participants reported in the form of a perceived lack of ability to understand, or motivation to adhere to, rules.

Our findings suggest that resistance and rebelliousness, as an overt form of rule-violation, may be becoming more observable–at least in regard to people's perceptions of non-adherence

in others—as resilience to coronavirus measures is increasingly tested. Indeed, resistance can be seen as a natural corollary of control particularly where such control is felt to be excessive or sustained [45]. Our findings reveal the ways in which some people may be performing 'resistance through rituals' [46], by for example eschewing or misappropriating new rituals (e.g. failing to wear masks or wearing them improperly) or by intentionally engaging in rituals that were once normative but which are in the context of a pandemic considered deviant (e.g. handshaking or not keeping socially distant). Although a form of violation, this rebelliousness might also be characterised as 'mis-behaviour' that 'arises from an impulse to take control rather than to be always subject of control . . . and seldom comes simply from a desire to break rules' *per se* [47].

Finally, our results suggest that further research is needed on the impact of vaccination programs on perceptions and behaviours related to adherence to measures. For some, the prospect of a vaccine might have an adverse impact on adherence, either by lowering the overall perception of risk in the general public. Research evidence on COVID-19 suggests that one of the key drivers of adherence in the subjective perception of risk [9–11]. However, for others, vaccine programs may not substantially change pre-existing attitudes to adherence. This theme, as with the others identified in our study, will warrant investigation in our ongoing research.

## Limitations

One limitation of this study is that it has likely overlooked a range of other factors that relate to (non-)adherence. As a qualitative and grounded analysis, we discuss only those most prominent themes that emerged from our particular data set. As discussed above, the nascent literature on COVID-19 policy adherence has found a range of other factors which have not been identified or discussed in the present study [9–12]. Also, although this study did not identify any patterns by demographic variables, this is potentially a result of relatively small sample size of a qualitative study with a diverse group of participants. It does not challenge the notion that life circumstances—for example, an individual's socio-economic status, age, geographic location (etc)—play an important role in adherence, and patterns reflecting this may be best identified by large sample quantitative surveys [5]. Another limitation of this study is that did not recruit participants from clinically extremely vulnerable or clinically vulnerable categories, for example, individuals aged 70 and over and those living with those with particular serious health conditions. Also, although our recruitment material did encourage those at high risk to apply, we received no applications from those over 70. This may be due to the fact that these are a hard-to-reach group online who are significantly less likely to use social media or be heavy internet users [48].

## Implications and recommendations for policy and practice

Further research is needed to explore possible approaches to promoting adherence (e.g. through incentives for adherence), and other potential factors impacting behaviours, such as situational events (e.g. public holidays), new threats and variants, and pharmaceutical developments such as vaccine rollouts. However, findings from our research provide insights that are helpful for policy development, and the introduction and implementation of various health security measures, educational campaigns, and health promotion activities during the current pandemic as well as future threats to public health. Our findings on alert fatigue, inconsistency in messaging, lack of trust in government and learned helplessness support our recommendations that government bodies and health officials should to consider ways to: (1) Strike a balance between open and transparent communication around measures and 'overloading' the public with information around rules and rule changes; (2) plan for measures that are as

unified as possible within and between countries in the UK and use as consistent a message (voice); (3) work to rebuild public trust, through exemplary adherence to rules amongst those in positions of authority. This study provides insights that are helpful for policy development, and the introduction and implementation of various health security measures, educational campaigns, and health promotion activities during the current pandemic as well as future threats to public health. Our findings suggest that latent and systemic failures—in the form of policy decisions that are commonly experienced as too changeable, inconsistent and confusing, and policy makers that are commonly perceived as untrustworthy–may play a significant role in creating the conditions that enable or encourage non-adherence.

## Supporting information

**S1 File. Focus group schedule.**
(DOCX)

## Author Contributions

**Conceptualization:** Simon N. Williams, Christopher J. Armitage, Tova Tampe, Kimberly A. Dienes.

**Data curation:** Simon N. Williams, Kimberly A. Dienes.

**Formal analysis:** Simon N. Williams, Kimberly A. Dienes.

**Funding acquisition:** Simon N. Williams, Christopher J. Armitage, Kimberly A. Dienes.

**Investigation:** Simon N. Williams, Christopher J. Armitage, Tova Tampe, Kimberly A. Dienes.

**Methodology:** Simon N. Williams, Christopher J. Armitage, Tova Tampe, Kimberly A. Dienes.

**Project administration:** Simon N. Williams, Kimberly A. Dienes.

**Resources:** Simon N. Williams, Kimberly A. Dienes.

**Software:** Simon N. Williams.

**Supervision:** Simon N. Williams.

**Validation:** Simon N. Williams, Kimberly A. Dienes.

**Writing – original draft:** Simon N. Williams.

**Writing – review & editing:** Simon N. Williams, Christopher J. Armitage, Tova Tampe, Kimberly A. Dienes.

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
