## [Decision Letter · Decision Letter 0]

27 Jul 2021

PONE-D-21-08563

Public perceptions of non-adherence to pandemic protection measures by self and others: a study of COVID-19 in the United Kingdom

PLOS ONE

Dear Dr. Williams,

Thank you for submitting your manuscript to PLOS ONE. After careful consideration, we feel that it has merit but does not fully meet PLOS ONE’s publication criteria as it currently stands. Therefore, we invite you to submit a revised version of the manuscript that addresses the points raised during the review process.

We look forward to receiving your revised manuscript.

Kind regards,

Maria Berghs, PhD

Academic Editor

PLOS ONE

Journal Requirements:

2. Please include additional information regarding the interview guide used in the study and ensure that you have provided sufficient details that others could replicate the analyses. For instance, if you developed an interview guide as part of this study and it is not under a copyright more restrictive than CC-BY, please include a copy, in both the original language and English, as Supporting Information."""

4. Thank you for stating the following in the Competing Interests/Financial Disclosure :

“CJA is supported by NIHR Manchester Biomedical Research Centre and NIHR Greater Manchester Patient Safety Translational Research Centre. TT is currently employed by the World Health Organization.”

We note that you received funding from a commercial source: NIHR Manchester Biomedical Research Centre and NIHR Greater Manchester Patient Safety Translational Research Centre

6. Thank you for submitting the above manuscript to PLOS ONE. During our internal evaluation of the manuscript, we found significant text overlap between your submission and the following previously published works, some of which you are an author.

https://www.researchgate.net/publication/232481491_Qualitative_Research_Theory_Method_and_Practice

Please revise the manuscript to rephrase the duplicated text, cite your sources, and provide details as to how the current manuscript advances on previous work. Please note that further consideration is dependent on the submission of a manuscript that addresses these concerns about the overlap in text with published work.

We will carefully review your manuscript upon resubmission, so please ensure that your revision is thorough

Reviewers' comments:

Reviewer's Responses to Questions

**Comments to the Author**

1. Is the manuscript technically sound, and do the data support the conclusions?

Reviewer #1: No

Reviewer #2: Yes

2. Has the statistical analysis been performed appropriately and rigorously? 

Reviewer #1: No

Reviewer #2: N/A

3. Have the authors made all data underlying the findings in their manuscript fully available?

Reviewer #1: No

Reviewer #2: No

4. Is the manuscript presented in an intelligible fashion and written in standard English?

Reviewer #1: Yes

Reviewer #2: Yes

5. Review Comments to the Author

Reviewer #1: This paper interviews 51 participants residing in the UK. They were guided to discuss the topics about the adherence to health behaviors on COVID-19 public attitudes of both their own and others. Analysis revealed six main reasons for non-adherence: 1. Alert fatigue, 2. Inconsistent rules 3. Lack of trust in government 4. Helplessness 5. Resistance and rebelliousness and 6. Reduced perception of risk and the prospect of vaccine.

This research did contribute to the qualitative expiration of the of the reasons for non-adherence. However, there still exists some weaknesses in this study.

In page 6, the author mentioned the study included participants who have experienced the mandated rules in Wales, Scotland, England and North Ireland. However, in this paper we don’t have a clear idea of that the mandates are and how many participants are involved in each. This information will lead to different conclusions of the analysis. For example, the second reason of the non-adherence is the inconsistent rules. Does all the policies in all these areas all inconstant from time to time?

In page 7, when introducing the participants and data collection, in line 163 the paper mentions that the participants are all recruited from online social media software, facebook and twitter. These samples are very bias as the author pointed out in the paper. Meanwhile in table demographic details are reported by the participants like gender, age and ethnicity. However, this paper doesn’t have any further introduction of any group characteristics associates with their attitudes of non-adherence.

When describing the reasons, the language is not precise and conclusive. It offers too many details of the words from each participant.

Typo: page 12, line 279

Reviewer #2: This paper describes a focus group study that examines reasons for non-adherence to required protective behaviours among the public during the pandemic. The topic is importance and will be of interest to many. The study is grounded in an up-to-date literature review. The rationale is logical. The design is appropriate including a relatively large sample size for a qual study. The analysis is conducted appropriately, and the paper is written clearly.

I had a number of small suggestions for improving the paper, mostly around providing further information. My main suggestion/ question is around the different protective behaviours. Handwashing, mask-wearing, distancing, and self-isolation are the main behaviours but they exist at different levels of difficulty and the reasons for (not) doing them migjt differ. But nowhere was this addressed or even mentioned in the paper (introduction, method – what were people asked they were (not)adhering in relation to, analysis – did the themes differ by behaviour?). This seems important to me, both for theory and for practice.

Other comments

Introduction

p. 10

Also, reported ‘complete adherence’ is

87 considerably higher than reported ‘majority adherence’ (where people are following most but

88 not all rules) [5] and

Can you clarify? You mean more people reported ‘completed adherence’ than reported ‘majority adherence’?

p. 10 However, a recent large international study concluded that

91 “pandemic policy fatigue” was widespread [7].

This sentence is not an accurate characterization of the findings of the Petherick paper. The data they analyse is adherence across time. The ‘fatigue’ term, in nearly all known usages, refers to tiredness of some sort, and is therefore a posited underlying variable. But there is no measure of the underlying variable, so references to the behaviour as ‘fatigue’ are circular. In addition, a simple prediction derived from ‘fatigue’ would be a linear decline in adherence. But a linear decline only occurs in some of the countries sampled. The overall shape is a u-shape. Why does adherence go up again? ‘Fatigue’ doesn’t explain this.

I couldn’t quite see how the Reason material fits it. How is it relevant to (non)adherence in the pandemic?

Method

The paper should be stand alone so the dis/advantages of online focus groups need to be stated tatyher than rferting yje reader to the aiythors’ previous papers.

p. 13 Participants in the final sample were recruited from across England, Wales,

161 Scotland and Northern Ireland, although specific numbers are not provided here to protect

162 anonymity.

How would mentioning how many came from each country compromise anonymity?

Why were volunteer groups approached?

We should see the wording in the topic guide. Adherence to what? Handwashing, masks, distancing and self-isolatiuon each have different levels of difficulty.

p. 15 Did the themes really ‘emerge’ from the transcripts? Some thematic analysis people (e.g. Braun & Clarke) criticize this formulation as it suggests that the researcher is passive.

What were they told on recruitment? (purpose of study?)

Results

What is the criteria for presentation?

The attribution bias is interesting.

p. 18-19

placing less attention on whatever the government is saying…. I’m not as concerned

about following the measures as seriously as I should have done.” (Participant 2,

259 Male, 30s)

Why is this quote partly in italics?

Reasons for non-adherence. It would be useful to know what behaviours are being referred to whether there were any differences by behaviours.

p. 21 The lack of respect for government, combined with the perception that rules were

345 inconsistent and unclear was a means of reconciling the cognitive dissonance (the mental

346 conflict arising from a mismatch of belief and behaviour) [31].

But did they experience ‘dissonance’? Is ‘feeling torn’ the evidence? Isn’t it just as likely to be an accountability concern that arises in the interaction rather than a cognitive mechanism?

I’d be interested in level of adherence too. Is this a group who are low in adherence?

Discussion

research on COVID-19 is starting to explore the adverse impact of poor

634 behavioural role modelling (the so-called “Cummings Effect” [4]).

The reference is to a critique of fatigue when it should be to the Lancet article of the same name by Fancourt.

6. PLOS authors have the option to publish the peer review history of their article (what does this mean?). If published, this will include your full peer review and any attached files.

Reviewer #1: No

Reviewer #2: **Yes: **John Drury

---

## [Author Response · Author response to Decision Letter 0]

11 Sep 2021

PLoS One rebuttal letter

Journal Requirements:

2. Please include additional information regarding the interview guide used in the study and ensure that you have provided sufficient details that others could replicate the analyses. For instance, if you developed an interview guide as part of this study and it is not under a copyright more restrictive than CC-BY, please include a copy, in both the original language and English, as Supporting Information."""

This has been included as Supporting Material (file S1)

4. Thank you for stating the following in the Competing Interests/Financial Disclosure :

“CJA is supported by NIHR Manchester Biomedical Research Centre and NIHR Greater Manchester Patient Safety Translational Research Centre. TT is currently employed by the World Health Organization.”

We note that you received funding from a commercial source: NIHR Manchester Biomedical Research Centre and NIHR Greater Manchester Patient Safety Translational Research Centre

We have added in the methods section:

“Ethical approval was received by Swansea University’s School of Management Research Ethics Committee. All participants gave informed consent, both written and verbal. All data were kept securely an confidentially in line with ethical requirements, and where data is presented below, all quotes are anonymised to protect participants’ identities.”

6. Thank you for submitting the above manuscript to PLOS ONE. During our internal evaluation of the manuscript, we found significant text overlap between your submission and the following previously published works, some of which you are an author.

https://www.researchgate.net/publication/232481491_Qualitative_Research_Theory_Method_and_Practice

We do cite the above work by Silverman – but only very briefly, in our analysis section (reference #25 - and using non-copied text. However, to ensure that it does not overlap- we have re-written this sentence as follows: “We also examined the data to look for information which might not ‘fit’ with the emerging themes (i.e. negative case analysis) the data [26]. These are discussed under ‘alternative accounts’ in the Results section.”)

Please revise the manuscript to rephrase the duplicated text, cite your sources, and provide details as to how the current manuscript advances on previous work. Please note that further consideration is dependent on the submission of a manuscript that addresses these concerns about the overlap in text with published work.

In terms of overlaps with our own work – it is likely that there are overlaps with the following Preprint of ours: https://www.medrxiv.org/content/10.1101/2020.11.17.20233486v1 (our understanding is that as a preprint – any duplicated text is acceptable) 

Also, some details of the methodology may be similar to our previous publications (which have used the same dataset and methods but to different ends and with different aims /sections 

However, we have re-written the methods section to ensure minimal overlap in wording etc, and to emphasise the differences in e.g. analysis, where they exist. 

We will carefully review your manuscript upon resubmission, so please ensure that your revision is thorough

Comments to the Author

Reviewer #1: This paper interviews 51 participants residing in the UK. They were guided to discuss the topics about the adherence to health behaviors on COVID-19 public attitudes of both their own and others. Analysis revealed six main reasons for non-adherence: 1. Alert fatigue, 2. Inconsistent rules 3. Lack of trust in government 4. Helplessness 5. Resistance and rebelliousness and 6. Reduced perception of risk and the prospect of vaccine.

This research did contribute to the qualitative expiration of the of the reasons for non-adherence. However, there still exists some weaknesses in this study.

In page 6, the author mentioned the study included participants who have experienced the mandated rules in Wales, Scotland, England and North Ireland. However, in this paper we don’t have a clear idea of that the mandates are and how many participants are involved in each. This information will lead to different conclusions of the analysis. For example, the second reason of the non-adherence is the inconsistent rules. Does all the policies in all these areas all inconstant from time to time?

We have emphasised the main changes in the Methods section, revising and extending the following text: “The study period included a number of important policy developments, which varied across the for UK nations, that provide a context for perceptions of government mandated rules including: the introduction of local lockdowns in Wales (introduced on 27 September); Scotland’s 16 day ban on drinking alcohol in licenses premises (introduced on 7th October); the ‘three tier’ system of COVID-19 restrictions in England (introduced on 12th October); the 4 week closure of pubs and restaurants in Northern Ireland (introduced on 14th October); the 19 day ‘firebreak’ lockdown in Wales (introduced 23rd October); Scotland’s five-tier system (introduced 21st October); and England’s four-week national lockdown (introduced 5th November). Beyond these major developments, a full list and description of all the specific policy changes, including how they differed across the four UK nations is beyond the scope of this research, but full details and ongoing updates of pandemic policies can be found on various ‘policy trackers’ (e.g. https://www.bsg.ox.ac.uk/research/research-projects/covid-19-government-response-tracker).”

Also, in the Results section (‘inconsistent rules’) – this was reiterated as follows:

“One of the main causes of this confusion was the perception that rules were inconsistent, either because they were changing so much over time (as discussed above as a cause of alert fatigue) or because they were inconsistent across place – that is between countries in the UK and between different regions within each country, due to the multiple policy changes noted above and in policy trackers [23]). Participants criticised what they saw as the inability, or unwillingness, of political leaders to create consistent policy and present a unified front across the different countries”

 the summary 

In page 7, when introducing the participants and data collection, in line 163 the paper mentions that the participants are all recruited from online social media software, facebook and twitter. These samples are very bias as the author pointed out in the paper. 

Thanks. Yes we do note this as a limitation in the original manuscript – but have now also added a sentence to emphasise this as a limitation in the discussion section: “…and might be more likely to be recruited via non-random sampling via social media”

Meanwhile in table demographic details are reported by the participants like gender, age and ethnicity. However, this paper doesn’t have any further introduction of any group characteristics associates with their attitudes of non-adherence.

We had a sentence in the original manuscript (‘Reasons for non-adherence, first paragraph) – but have now emphasised it. On the new manuscript, it reads: “We did not find any obvious patterns or differences according to the demographics of the participants, with a mix of genders, ages and races and ethnicities being represented in each theme. As noted above, regardless of their demographics, all participants reported generally high self- adherence.” 

When describing the reasons, the language is not precise and conclusive. It offers too many details of the words from each participant.

Typo: page 12, line 279

Thanks- I think this may have referred to the per se which has now been italicized to per se (we couldn’t see any other typo’s at this point, but have proof read the manuscript and made corrections throughout).

Reviewer #2: This paper describes a focus group study that examines reasons for non-adherence to required protective behaviours among the public during the pandemic. The topic is importance and will be of interest to many. The study is grounded in an up-to-date literature review. The rationale is logical. The design is appropriate including a relatively large sample size for a qual study. The analysis is conducted appropriately, and the paper is written clearly.

Thank you for your positive comments

I had a number of small suggestions for improving the paper, mostly around providing further information. My main suggestion/ question is around the different protective behaviours. Handwashing, mask-wearing, distancing, and self-isolation are the main behaviours but they exist at different levels of difficulty and the reasons for (not) doing them migjt differ. But nowhere was this addressed or even mentioned in the paper (introduction, method – what were people asked they were (not)adhering in relation to, analysis – did the themes differ by behaviour?). This seems important to me, both for theory and for practice.

Thanks for this important suggestion. The study, and focus group discussions were primarily focused on government mandated measures – and particularly (but not exclusively) ‘social distancing’ measures (and less emphasise on infection-reducing behaviours and guidance, e.g. hand hygiene etc). We have tried to add in more detail about this focus. We have also included a supplementary file showing the focus group schedule. Due to the semi-structured nature of the focus groups, participants were asked to define what they understood as ‘the rules’ (and the extent to which they were following them) and also what ‘social distancing’ meant to them. They were prompted where necessary with specific types of SD – e.g 2 metre rule. The following text has been added to hopefully aid clarity and precision:

Introduction

“During the pandemic, in the UK, the social distancing measures that are the focus of this study included, but were not limited to: keeping physically separate (1-2 meters in the UK), only meeting with others in ways permitted under current legislation, and self-isolating when required to do so”

“We focus on adherence to government mandates and regulation – particularly including social distancing measures such as keeping 1-2 metres apart from others, only meeting and mixing with others where and when permitted, but also including other mandated measures such as facemask requirements.”

Methods

“Focus groups were semi-structured and contained a lengthy section on adherence, particularly focused on adherence to government mandates and legal and official measures around social distancing (as noted above, including keeping 1-2 metres physically distant, only meeting and mixing with others as and when permitted). In discussing social distancing measures, participants at times voluntarily discussed (non-)adherence to other rules, including mask mandates. The main topics for the focus groups included: what people thought about their own adherence and the adherence of others to COVID-19 social distancing measures, what they thought the reasons for adherence or non-adherence were, as well as broader topics related to people’s views and experiences around any impact of the pandemic on work life, social life and mental health – the focus of previous and future publications from the project. The focus group topic guide is included as Supporting Material (S1).”

Other comments

Introduction

p. 10

Also, reported ‘complete adherence’ is

87 considerably higher than reported ‘majority adherence’ (where people are following most but

88 not all rules) [5] and

Can you clarify? You mean more people reported ‘completed adherence’ than reported ‘majority adherence’?

Thanks for catching this. We had included a type-error – whereby complete adherence (as one might expect) is much lower than majority adherence – we have also clarified the definition of complete compliance. 

p. 10 However, a recent large international study concluded that

91 “pandemic policy fatigue” was widespread [7].

This sentence is not an accurate characterization of the findings of the Petherick paper. The data they analyse is adherence across time. The ‘fatigue’ term, in nearly all known usages, refers to tiredness of some sort, and is therefore a posited underlying variable. But there is no measure of the underlying variable, so references to the behaviour as ‘fatigue’ are circular. In addition, a simple prediction derived from ‘fatigue’ would be a linear decline in adherence. But a linear decline only occurs in some of the countries sampled. The overall shape is a u-shape. Why does adherence go up again? ‘Fatigue’ doesn’t explain this.

Thanks for the really helpful comments. We have rewritten how we present the findings of the Petherick paper – including some of your comments above. It now reads:

“However, a recent international study posited that ‘fatigue’ might explain an observed decline in adherence for ‘high-cost’ behaviours like social distancing (but not for ‘low-cost’ ones like mask-wearing). For high and middle income countries this decline was reversed later in the pandemic [7]. Further behavioural research is needed to add to our understanding of the potential existence and role of “fatigue” during the pandemic [8].”

I couldn’t quite see how the Reason material fits it. How is it relevant to (non)adherence in the pandemic?

Method

The paper should be stand alone so the dis/advantages of online focus groups need to be stated tatyher than rferting yje reader to the aiythors’ previous papers.

Thanks – we have added the following details: “Online focus groups were a necessary requirement of the pandemic, but have previously been shown to have the advantage of conveniently obtaining the views of a diverse group of geographically disparate group of individuals (although they potentially preclude those without the requisite communication technology)”

p. 13 Participants in the final sample were recruited from across England, Wales,

161 Scotland and Northern Ireland, although specific numbers are not provided here to protect

162 anonymity.

How would mentioning how many came from each country compromise anonymity?

These have been added to table 1, as has the in-text note: “the final sample was primarily from England and Wales (primarily an effect of the snowball sampling technique used).”

Why were volunteer groups approached?

The following has been added, to explain the rationale for adding community and volunteer groups: “Volunteering and community sites were chosen due to their common use as sites to advertise research studies (although this increases the potential risk of a biased sample- toward those who may be more inclined to take part in research studies). 

We should see the wording in the topic guide. Adherence to what? Handwashing, masks, distancing and self-isolatiuon each have different levels of difficulty.

This has been included as supplemental file 1 (S1). As noted above, this study focused particularly on adherence to ‘social distancing’ measures (rather than on other infection-reducing behaviour and guidance, such as mask wearing). Participants adherence specially focused on government mandated measures – and particularly (but not exclusively) ‘social distancing’ measures (participants were asked to define what they thought ‘social distancing’ and ‘the rules’ were and the extent to which they adhered to them. 

p. 15 Did the themes really ‘emerge’ from the transcripts? Some thematic analysis people (e.g. Braun & Clarke) criticize this formulation as it suggests that the researcher is passive.

Thanks – this has been (we agree) amended to “Data collection and analysis continued until no new significant themes were developed.”

What were they told on recruitment? (purpose of study?)

The following has been added: “As part of the initial recruitment, participants were told the study was looking at the UK public’s perceptions and experiences of social isolation and social distancing measures during the pandemic.”

Results

What is the criteria for presentation?

The attribution bias is interesting.

p. 18-19

placing less attention on whatever the government is saying…. I’m not as concerned

about following the measures as seriously as I should have done.” (Participant 2,

259 Male, 30s)

Why is this quote partly in italics?

Reasons for non-adherence. It would be useful to know what behaviours are being referred to whether there were any differences by behaviours.

p. 21 The lack of respect for government, combined with the perception that rules were

345 inconsistent and unclear was a means of reconciling the cognitive dissonance (the mental

346 conflict arising from a mismatch of belief and behaviour) [31].

But did they experience ‘dissonance’? Is ‘feeling torn’ the evidence? Isn’t it just as likely to be an accountability concern that arises in the interaction rather than a cognitive mechanism?

I’d be interested in level of adherence too. Is this a group who are low in adherence?

We agree that cognitive dissonance was not the most appropriate way to understand this finding – and have expanded our discussion of the section on trust to include your suggestion that it is related to the issue of accountability :

“Some participants argued that the lack of respect for government was being used by some to account for engaging in forms of social mixing that was not permitted within the rules. …” 

Discussion

research on COVID-19 is starting to explore the adverse impact of poor

634 behavioural role modelling (the so-called “Cummings Effect” [4]).

The reference is to a critique of fatigue when it should be to the Lancet article of the same name by Fancourt.

Thanks for pointing this out – this was a type-o, it should have read [40] instead of [4] – and has now been changed

DECLARATIONS 

Competing interest statement: CJA is supported by NIHR Manchester Biomedical Research Centre and NIHR Greater Manchester Patient Safety Translational Research Centre. TT is currently employed by the World Health Organization. This does not alter our adherence to PLOS ONE policies on sharing data and materials.” (as detailed online in our guide for authors http://journals.plos.org/plosone/s/competing-interests). The authors have no other relationships or activities that could appear to have influenced the submitted work. 

Transparency declaration: The lead author (the manuscript’s guarantor) affirms that the manuscript is an honest, accurate, and transparent account of the study being reported; that no important aspects of the study have been omitted; and that any discrepancies from the study as planned (and, if relevant, registered) have been explained. 

Authors’ contributions: All authors contributed to the planning of the study. The analysis was conducted by SW and KD. The initial draft of the article was written by SW. All authors revised the manuscript and approved the final version for publication. SW is the guarantor of the article. 

Funding statement: This research was supported by the Manchester Centre for Health Psychology based at the University of Manchester (£2000) and Swansea University’s ‘Greatest Need Fund’ (£3000). 

Data availability statement: Although the authors cannot make their study’s data publicly available at the time of publication, all authors commit to make the data underlying the findings described in this study fully available without restriction to those who request the data, in compliance with the PLOS Data Availability policy. For data sets involving personally identifiable information or other sensitive data, data sharing is contingent on the data being handled appropriately by the data requester and in accordance with all applicable local requirements. 

Ethical restrictions related to participant confidentiality, as agreed during ethical review, prohibit the authors from making the data set publicly available. During the consent process, participants were explicitly guaranteed that the data would only be seen my members of the study team, including the authors of the current paper and any research assistants as approved via the ethical review. Data is kept securely and confidentially and anonymously on password-protected files for a minimum of ten years as per the ethical review requirements. Requests for access to data will be considered on a case-by-case basis, and data will be made available provided: a) any data made available does not compromise the identity or confidentiality of any participants and b) particularly where there are public health grounds for the release of data. For any discussions about the data set please contact the relevant members of the research team Dr Simon Williams (s.n.williams@swansea.ac.uk) or Dr Kimberly Dienes k.dienes@swansea.ac.uk . 

Ethics statement: Ethical approval was received by Swansea University’s School of Management Research Ethics Committee.

---

## [Decision Letter · Decision Letter 1]

6 Oct 2021

Public perceptions of non-adherence to pandemic protection measures by self and others: a study of COVID-19 in the United Kingdom

PONE-D-21-08563R1

Dear Dr. Williams,

We’re pleased to inform you that your manuscript has been judged scientifically suitable for publication and will be formally accepted for publication once it meets all outstanding technical requirements.

Kind regards,

Maria Berghs, PhD

Academic Editor

PLOS ONE

Additional Editor Comments (optional):

Thank-you for your patience with this manuscript and the time it has taken to go through the peer-review process. There is just a minor remark by one of the reviewers but the paper is accepted.

Reviewers' comments:

Reviewer's Responses to Questions

**Comments to the Author**

1. If the authors have adequately addressed your comments raised in a previous round of review and you feel that this manuscript is now acceptable for publication, you may indicate that here to bypass the “Comments to the Author” section, enter your conflict of interest statement in the “Confidential to Editor” section, and submit your "Accept" recommendation.

Reviewer #2: All comments have been addressed

2. Is the manuscript technically sound, and do the data support the conclusions?

Reviewer #2: Yes

3. Has the statistical analysis been performed appropriately and rigorously? 

Reviewer #2: N/A

4. Have the authors made all data underlying the findings in their manuscript fully available?

Reviewer #2: Yes

5. Is the manuscript presented in an intelligible fashion and written in standard English?

Reviewer #2: Yes

6. Review Comments to the Author

Reviewer #2: It would have been useful to include a statement such as 'the extracts presented are representative of the sample', as I suggested.

7. PLOS authors have the option to publish the peer review history of their article (what does this mean?). If published, this will include your full peer review and any attached files.

Reviewer #2: **Yes: **John Drury

---

## [Editor Report · Acceptance letter]

14 Oct 2021

PONE-D-21-08563R1 

Public perceptions of non-adherence to pandemic protection measures by self and others: a study of COVID-19 in the United Kingdom 

Dear Dr. Williams:

I'm pleased to inform you that your manuscript has been deemed suitable for publication in PLOS ONE. Congratulations! Your manuscript is now with our production department. 

Kind regards, 

on behalf of

Dr. Maria Berghs 

Academic Editor

PLOS ONE